# Micronutrient dynamics and deficiency risk across pregnancy and postpartum in a Slovak cohort

Alexandra Kristufkova[1⊗], Neha Basheer [2⊗], Katarina Koprdova[3], Matus Lieskovsky[4], Michal Fresser[4], Norbert Zilka[2]*

1  1st Department of Obstetrics and Gynecology, St. Cyril and Method's Hospital, Faculty of Medicine, Comenius University, Bratislava, Slovak Republic, 2  Institute of Neuroimmunology of Slovak Academy of Sciences, Bratislava, Slovak Republic, 3  Department of Obstetrics and Gynecology, Bory - Penta Hospitals, Bratislava, Slovak Republic, 4  Mumo Health j. s. a., Bratislava, Slovak Republic

⊗ These authors contributed equally to this work.
* norbert.zilka@savba.sk

## Abstract

### Objective

To assess the dynamics in blood concentrations of vitamins (A, B6, B12, D, E,), trace elements such as selenium, magnesium, zinc, and iron (transferrin), and metabolite homocysteine during pregnancy and postpartum.

### Design

Cross-sectional, national cohort study conducted between January and June 2024.

### Setting

Slovakia.

### Population

Pregnant and postpartum women.

### Methods

From venous blood and capillary dry blood spot micronutrients were analysed using standard biochemical and biophysical methods.

### Main outcome measures

Group differences in blood micronutrient levels across pregnancy and postpartum.

### Results

Our findings reveal significant differences in maternal micronutrient levels across pregnancy and postpartum. While some nutrients, including vitamin D and folate, remained

**Data availability statement:** All relevant data are contained within the manuscript. The dataset is fully available, without restriction, from the time of publication at Dryad (DOI: 10.5061/dryad.8w9ghx409).

**Funding:** We thank Mumo Health for their generous support, which contributed to the execution of this study. The funders had no role in study design, data collection and analysis, decision to publish, or preparation of the manuscript.

**Competing interests:** The authors have declared that no competing interests exist.

relatively stable, others such as vitamin A, B12, iron and zinc were observed at lower levels, and vitamin E at higher levels during pregnancy. Vitamin E levels in the 3rd trimester frequently exceeded reference values for the general adult population, whereas zinc levels were significantly lower postpartum. We observed high prevalence of vitamin B12 and iron deficiencies, as indicated by transferrin saturation, particularly in the 3rd trimester. Vitamin D deficiency was prevalent throughout pregnancy and postpartum. Finally, our analysis demonstrated that dried blood spot (DBS) technology provides comparable results to venous blood analysis for measuring vitamin A, D and homocysteine levels.

## Introduction

Nutrient deficiency during pregnancy is a well-documented issue in developing countries, primarily due to inadequate access to a well-balanced diet. In Europe, previous studies have indicated that pregnant women are most commonly deficient in vitamin B9, D and iron [1–6]. However, comprehensive data on nutrient deficiencies across all European countries remain limited. Moreover, trimester-specific reference ranges for key nutrients are still lacking, making it difficult to assess and address deficiencies at different stages of pregnancy with precision.

Recent findings from the NiPPeR trial – a multicenter, double-blind, randomized controlled study conducted in the UK, New Zealand, and Singapore – revealed that over 90% of participants (1,729 women aged 18–38 years) had marginal or low levels of one or more essential vitamins (vitamin B2, B9, B12, and vitamin D) prior to conception. Additionally, many developed a vitamin B6 deficiency in late pregnancy [7]. These results highlight that nutrient deficiencies are highly prevalent among pregnant women, even in high-income countries.

These findings suggest that nutrient deficiencies among pregnant women in Europe may be more widespread than previously assumed. Given the absence of routine monitoring for blood nutrient concentrations, national studies remain the only available method to assess the extent of this issue. In our study, we analysed a Slovak population of pregnant and postpartum women, tracking vitamin A, B9 (folate), B12, and D; and trace elements such as selenium (Se), magnesium (Mg), iron (as transferrin), zinc (Zn) and homocysteine (HYC) fluctuations throughout pregnancy and the postpartum period. To address the need for improved clinical monitoring, we evaluated the feasibility of dried blood spot (DBS) technology for nutrient analysis.

## Materials and methods

### Ethical considerations

The study protocol was approved by the Ethics Committee of University Hospital in Bratislava at St. Cyril and Methodius Hospital on 23rd March 2023, ensuring compliance with ethical guidelines. The study was registered under registration number EK 1/3/2023, and all participants provided written informed consent before enrolment.

## Study design and participants

This cross-sectional study included women with both spontaneously conceived and ART-assisted pregnancies, verified through ultrasound confirmation of a viable singleton intrauterine pregnancy. The initial recruitment target was 150 participants, with final enrolment at 127 due to exclusion for non-compliance with inclusion criteria or withdrawal of consent. Eligibility criteria required a confirmed singleton pregnancy (to control for differences in fetal nutrient demands), while exclusion criteria included preexisting metabolic disorders such as diabetes (to avoid confounding effects on metabolism), and prior micronutrient supplementation (to ensure baseline comparability).

## Sample collection and outcome measure

Participants were recruited from the 1st department of Obstetrics and Gynaecology of Faculty of medicine Comenius University and University Hospital in Bratislava between 1st January and 30th June 2024. At the initial prenatal visit, participants completed a structured, physician-administered medical history questionnaire, followed by a self-reported questionnaire postpartum. Venous blood samples (EDTA plasma and serum) were collected following an overnight fast at three standardized time points: 1st trimester (11–12 weeks) for baseline micronutrient profiling and supplementation guidance; 2nd trimester (24–28 weeks, coinciding with the oral glucose tolerance test) for reassessment; and 3rd trimester (34–36 weeks) for final monitoring. Samples were stored at –80 °C until analysis. Additionally, participants were instructed to perform a self-administered dried blood spot (DBS) test using the Mumo Health kit (Bratislava, Slovakia), which involved capillary blood collection via finger prick per the manufacturer's protocol. DBS samples were stored at 4 °C until analysis.

Micronutrient concentrations were quantified using standardized clinical assays. From venous blood, vitamins A and E were measured by high-performance liquid chromatography (HPLC), while magnesium, homocysteine (HCY), and transferrin (TRF) were quantified via photometry using the Siemens Advia platform. Folate (vitamin B9) was assessed using chemiluminescent immunoassay (CLIA) on the Siemens Centaur system. Vitamins B12 and D were measured by electrochemiluminescence immunoassay (ECLIA) on the Roche Cobas analyzer. Zinc concentrations were determined using atomic absorption spectrometry with electrothermal atomization (AAS-ET; Agilent), and selenium was quantified via inductively coupled plasma mass spectrometry (ICP-MS). All venous blood analyses were performed at Medirex Diagnostics (Bratislava, Slovakia). Dried blood spot (DBS) samples were analysed by Vitas Analytical Services (Oslo, Norway); trace elements were measured using ICP-MS, and vitamin concentrations were assessed using tandem quadrupole mass spectrometry.

## Statistical analysis

Statistical analyses were conducted to evaluate micronutrient correlations and group differences. Supplement use, BMI, and parity were excluded as covariates due to missing or inconsistent data across groups, substantial variability in supplement intake, and challenges in standardizing BMI during pregnancy. Spearman partial correlation coefficients were calculated to assess relationships between micronutrients, adjusting for age; statistical significance was tested using the $\chi^2$ test, and p-values were corrected for multiple comparisons using the Holm method. Group differences across trimesters, postpartum, and age-matched non-pregnant controls were assessed using simple linear regression models. To account for potential heteroskedasticity, a heteroskedasticity-consistent (robust) variance estimator was applied. Overall significance was evaluated using Fisher's F-test ($\alpha = 0.05$). For pairwise group comparisons, post hoc Student's t-tests were performed with Holm-adjusted p-values to control for multiple testing.

# Results

## Study population

A total of 127 women including healthy age matched non-pregnant controls (n = 25), pregnant participants in their first (n = 18), second (n = 21), and third trimesters (n = 39) and postpartum women (n = 24) aged 21–47 years, were enrolled. The mean age was 33.20 ± 7.55 years in the nonpregnant (NP) control group, 32.94 ± 5.22 years in the first trimester (1st)

group, 32.95±5.34 years in the second trimester (2nd) group, 32.92±5.00 years in the third trimester (3rd) group, and 32.00±4.09 years in the postpartum (PP) group. There were no statistically significant differences in age across groups (p > 0.05).

## Stage-dependent variability in vitamins, trace elements, and homocysteine levels during pregnancy and postpartum

Group-wise comparisons revealed significant stage-dependent alterations in several micronutrients during pregnancy and postpartum (Table 1). Vitamin A (retinol) levels were significantly lower across all trimesters compared to NP (1st and 2nd: p = 0.02; 3rd: p < 0.001), with higher levels observed in PP versus the 3rd trimester (p = 0.02), although levels remained below NP (Fig 1a). Folate and vitamin D levels remained stable across all groups (p = 0.10 and p = 0.36, respectively) (Fig 1b, d). Vitamin B12 levels were significantly lower in the 3rd trimester relative to NP (p < 0.001), with

**Table 1. Venous blood concentrations of vitamins, trace elements and homocysteine across pregnancy and postpartum.**

| Stage | Age | A µmol/l | B9 nmol/l | B12 pmol/l | D µg/l | E µmol/l | Mg mmol/l | Zn µmol/l | Se mol/l | TRF g/l | HCY µmol/l |
|---|---|---|---|---|---|---|---|---|---|---|---|
| **Range** | 21−45 | <0.35** 0.35−0.65* 1.05−2.1 >4.9## | <7.60* 7.60−12.20 >12.20# | 165−672 | <20* 21−29^ 30−44 >60# | 11.6−42.0 | 0.53−1.11 | 11.0−23.0 | 0.75−1.86 | 2.50−3.80 | 5.0−15.0 |
| **Mean** | | | | | | | | | | | |
| NP | 32.28 | 2.26 | 19.21 | 235.20 | 25.49 | 33.74 | 0.82 | 14.23 | 0.98 | 2.89 | 11.97 |
| 1st | 32.94 | 1.78 | 30.24 | 221.67 | 24.88 | 32.62 | 0.76 | 13.64 | 0.93 | 3.02 | 6.93 |
| 2nd | 32.95 | 1.76 | 24.20 | 187.86 | 25.92 | 40.84 | 0.72 | 12.38 | 0.87 | 3.62 | 6.91 |
| 3rd | 33.00 | 1.51 | 26.43 | 169.82 | 29.44 | 47.48 | 0.76 | 12.72 | 0.81 | 4.02 | 8.24 |
| PP | 35.00 | 1.93 | 22.51 | 179.62 | 26.76 | 46.86 | 0.74 | 9.25 | 0.87 | 3.51 | 8.63 |
| **SD** | | | | | | | | | | | |
| NP | 7.19 | 0.65 | 12.38 | 75.43 | 8.30 | 7.43 | 0.05 | 3.51 | 0.17 | 0.50 | 7.38 |
| 1st | 5.22 | 0.42 | 14.40 | 66.26 | 8.46 | 4.71 | 0.06 | 3.27 | 0.15 | 0.42 | 2.21 |
| 2nd | 5.34 | 0.46 | 15.25 | 62.92 | 8.55 | 7.95 | 0.05 | 2.87 | 0.17 | 0.57 | 2.21 |
| 3rd | 4.96 | 0.50 | 15.60 | 53.43 | 9.88 | 10.45 | 0.07 | 4.10 | 0.10 | 0.68 | 3.05 |
| PP | 4.09 | 0.59 | 11.71 | 81.28 | 14.02 | 12.20 | 0.06 | 3.28 | 0.19 | 0.56 | 2.32 |
| **Min.** | | | | | | | | | | | |
| NP | 21.00 | 1.19 | 5.82 | 129.00 | 8.82 | 21.35 | 0.72 | 4.94 | 0.59 | 1.72 | 7.20 |
| 1st | 26.00 | 1.04 | 9.28 | 120.00 | 11.80 | 23.47 | 0.64 | 7.58 | 0.67 | 2.14 | 3.80 |
| 2nd | 23.00 | 0.97 | 3.45 | 94.00 | 9.20 | 25.92 | 0.64 | 8.97 | 0.67 | 2.43 | 3.70 |
| 3rd | 25.00 | 0.74 | 4.30 | 80.00 | 8.75 | 28.49 | 0.66 | 6.54 | 0.59 | 3.02 | 3.20 |
| PP | 28.00 | 1.08 | 5.93 | 86.00 | 6.06 | 30.47 | 0.66 | 3.23 | 0.55 | 2.40 | 5.80 |
| **Max.** | | | | | | | | | | | |
| NP | 45.00 | 3.48 | 50.44 | 397.00 | 43.50 | 54.86 | 0.91 | 22.04 | 1.28 | 3.80 | 45.50 |
| 1st | 44.00 | 2.46 | 54.36 | 358.00 | 43.30 | 39.09 | 0.88 | 21.27 | 1.33 | 3.69 | 12.10 |
| 2nd | 41.00 | 2.93 | 54.36 | 321.00 | 43.70 | 53.46 | 0.84 | 19.95 | 1.33 | 5.05 | 12.90 |
| 3rd | 43.00 | 2.56 | 54.36 | 373.00 | 51.40 | 74.49 | 1.02 | 27.94 | 1.08 | 5.81 | 16.40 |
| PP | 43.00 | 3.28 | 50.47 | 384.00 | 56.20 | 79.08 | 0.90 | 21.27 | 1.44 | 4.86 | 14.40 |

Values represent mean, standard deviation (SD), minimum, and maximum venous blood concentrations for each time point: non-pregnant (NP), trimester 1 (1st), trimester 2 (2nd), trimester 3 (3rd), and postpartum (PP). Reference ranges are shown above each column.

*\*\*Deficiency; \*Manifests deficiency; ^Insufficient; #Higher concentration; ##Toxic. Unless otherwise indicated, ranges represent upper and lower limits of normative values based on the analytical platform used.*

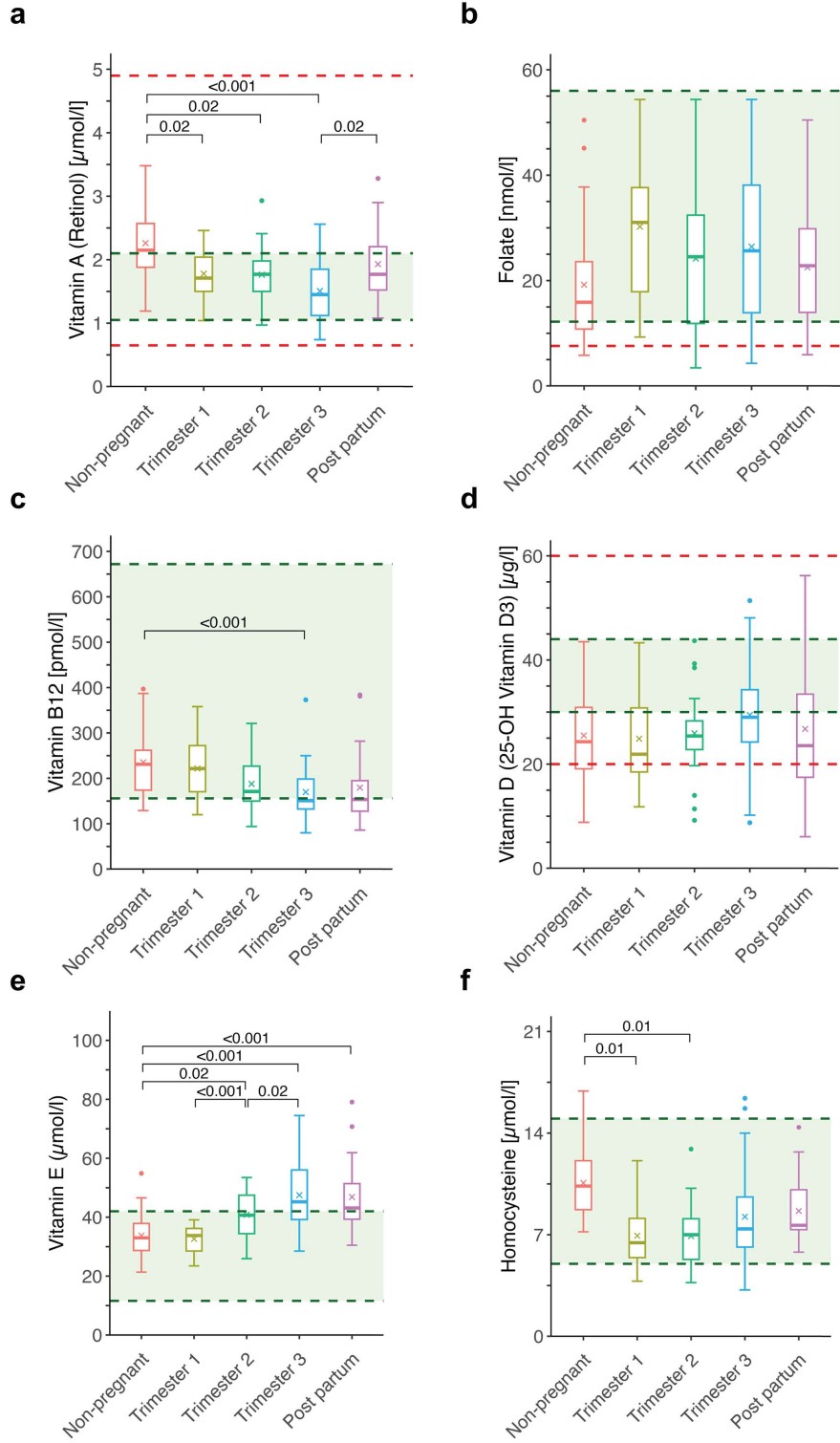

**Fig 1. Longitudinal changes in venous blood concentrations of micronutrients and homocysteine across pregnancy and postpartum.** Box-plots show levels of (a) vitamin A (retinol), (b) folate (vitamin B9), (c) vitamin B12, (d) vitamin D (25-hydroxyvitamin **D)**, (e) vitamin E and (f) homocys-teine in venous blood across five time points: non-pregnant, trimester 1, trimester 2, trimester 3, and postpartum. Dashed red lines indicate deficiency

or extensive levels (*vitamin A* 0.65 µmol/L deficiency, > 4.9 µmol/L toxicity; *folate* < 7.60 nmol/L deficiency; *vitamin D* < 20 µg/L deficiency, > 60 µg/L extensive levels); green shading denotes the normal range (*vitamin A* 1.05–2.1 µmol/L; *folate* > 12,20 nmol/L; *vitamin B12* 156–672 pmol/L; *vitamin D* 30–44 µg/L; *vitamin E* 11.6–42 µmol/L; *homocysteine* 5.0–15.0 µmol/L). Boxes represent the interquartile range (IQR), horizontal lines denote medians, and whiskers extend to 1.5 × IQR.

PP values also lower but not statistically different (p = 0.10) (Fig 1c). Vitamin E levels were significantly higher beginning in the 2nd trimester (p = 0.02), with further elevation in the 3rd trimester (p < 0.001) and postpartum (p < 0.001); no difference was observed in the 1st trimester (p = 0.98) (Fig 1e). Homocysteine levels were significantly lower in the 1st and 2nd trimesters relative to NP (both p = 0.01), with similar non-significant differences persisting into the 3rd trimester and postpartum (Fig 1f).

Magnesium levels were consistently higher in NP women than in all pregnancy and postpartum groups (all p ≤ 0.01), though differences between pregnancy stages were not significant (Fig 2a). In contrast, transferrin levels were significantly higher in the 2nd trimester compared to NP women (p < 0.001), remained elevated in the 3rd trimester (p < 0.001), and were lower in PP, though still above NP levels (p < 0.001). All consecutive group comparisons were significant (p ≤ 0.04) (Fig 2b). Zinc levels remained unchanged across pregnancy but were significantly lower in PP, both in comparison to NP (p < 0.001) and to the 3rd trimester (p < 0.001) (Fig 2c). Selenium levels were significantly lower only in the 3rd trimester compared to NP (p < 0.001), with no other intergroup differences observed (Fig 2d).

## Micronutrient correlation network reveals targeted associations

Correlation analysis identified several significant associations between micronutrients. A moderate negative correlation was observed between folate and homocysteine (r = −0.44, 95% CI: −0.64 to −0.17, p < 0.001), while folate also showed a moderate positive correlation with vitamin D (r = 0.40, 95% CI: 0.12 to 0.61, p < 0.001). A strong positive correlation was detected between transferrin and vitamin E (r = 0.49, 95% CI: 0.24 to 0.68, p < 0.001). Additionally, low but significant positive correlations were found between selenium and vitamin B12 (r = 0.33, 95% CI: 0.05 to 0.57, p = 0.01), and between magnesium and vitamin B12 (r = 0.38, 95% CI: 0.09 to 0.59, p < 0.001). No other pairwise correlations reached statistical significance after multiple comparison correction (Fig 3).

## High prevalence of micronutrient deficiencies during pregnancy and postpartum

Assessment of micronutrient status revealed a higher prevalence in zinc deficiency (< 11 µmol/l) across pregnancy, rising from 16.7% in NP women to 42.9% in the 2nd trimester, and peaking at 87.5% PP. Vitamin B12 deficiency (< 156 pmol/l) also more frequently observed, from 20.8% in NP women to 28.6% in the 2nd trimester and 56.4% in the 3rd trimester, remaining elevated PP (50%). Vitamin D deficiency (< 20 µg/l) was present in 37.5% of NP women, decreasing to 15.4% by the 3rd trimester; however, insufficient levels (21–29 µg/l) remained common across pregnancy, affecting up to 57.1% of women. One PP participant exhibited elevated vitamin D levels exceeding the upper threshold (> 60 µg/l).

Vitamin A insufficiency (< 1.05 µmol/l) was identified in 5.6% of women during the 1st trimester and 17.9% during the 3rd trimester. Elevated vitamin E levels (>42 µmol/L) were observed in 42.9% and 59.0% of participants in the 2nd and 3rd trimesters, respectively, compared to 8.3% of NP women; elevated levels remained prevalent postpartum (50%). Homocysteine (> 15 µmol/l) was rarely elevated during pregnancy or postpartum. Transferrin elevation (> 3.8 g/l) increased throughout pregnancy (33.3% in the 2nd trimester; 53.9% in the 3rd), before declining PP (16.7%). Conversely, low transferrin levels (< 2.5 g/l) were present in 16.7% of NP women and 4.8–5.6% in early pregnancy. Selenium deficiency (< 0.75 mol/l) was observed in 12.5% of NP women and in 25.6% by the 3rd trimester (Table 2).

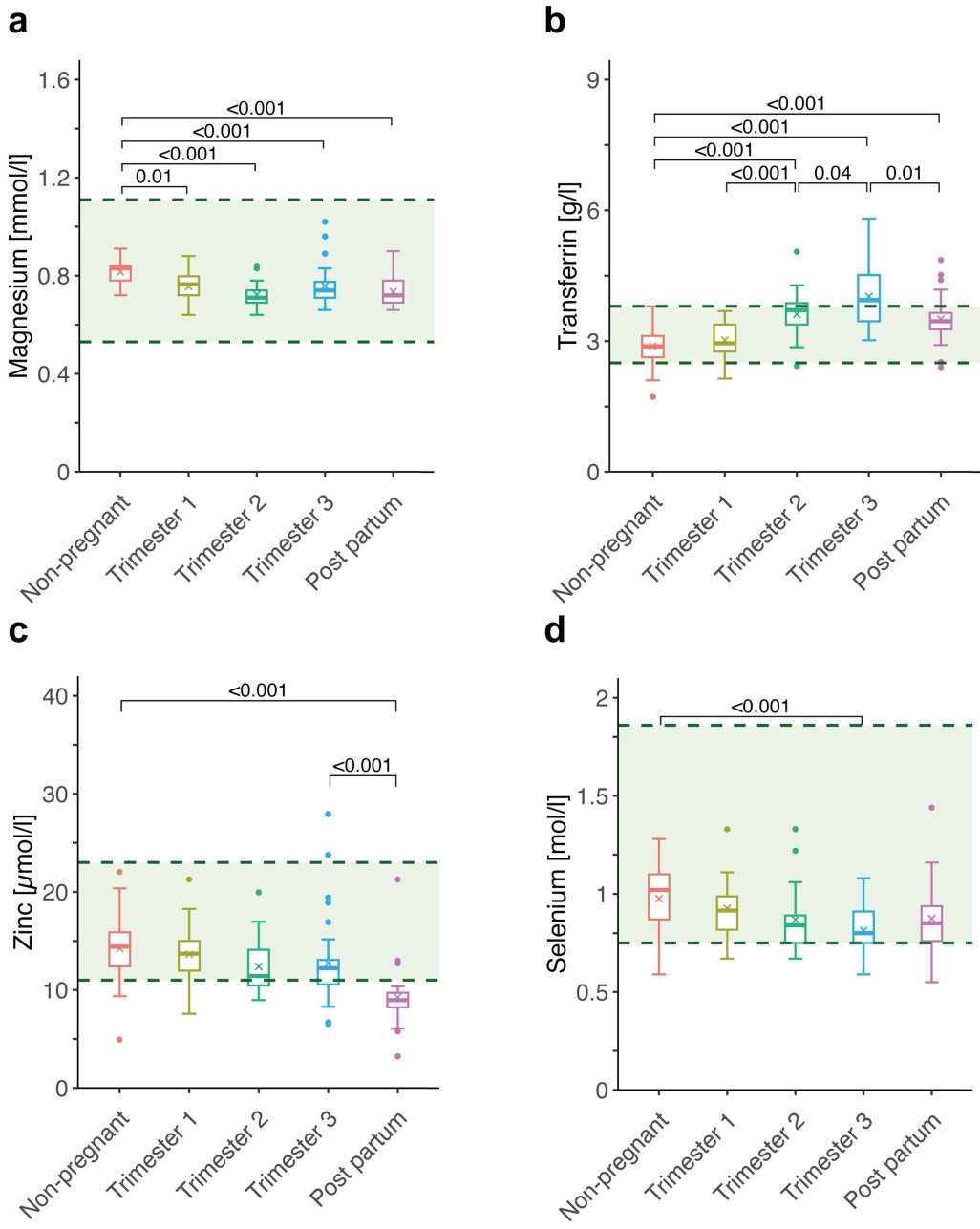

**Fig 2. Longitudinal changes in venous blood concentrations of trace elements across pregnancy and postpartum.** Boxplots show levels of (a) magnesium, (b) transferrin, (c) zinc and (d) selenium in venous blood at five time points: non-pregnant, trimester 1, trimester 2, trimester 3, and postpartum. Green-shaded areas represent reference ranges (*magnesium* 0.53–1.11 mmol/L; *transferrin* 5.0–15.0 µmol/L; *zinc* 11.0–23.0 µmol/L; *selenium* 0.75–1.86 µmol/L). Boxes show interquartile ranges; medians are marked by horizontal lines and whiskers extend to 1.5×IQR.

## Dried blood spot measurements yield high concordance with venous blood for vitamin A, vitamin D, and homocysteine

To facilitate at-home sample collection in pregnant participants, where minimally invasive procedures are preferable, we selected dried blood spot (DBS) sampling as an alternative to venous blood collection. We selected three analytes for

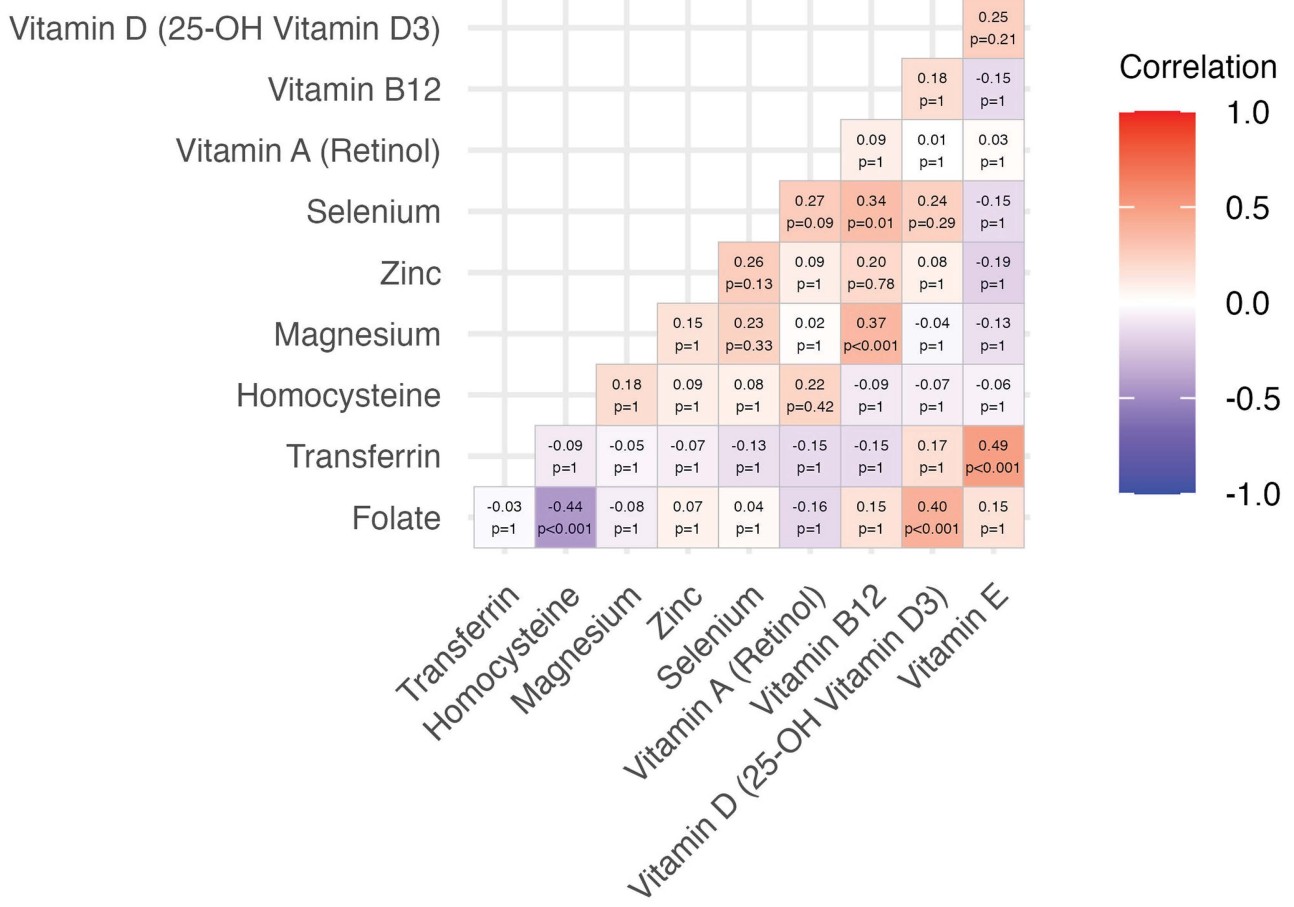

**Fig 3. Spearman partial correlations between micronutrients.** Spearman partial correlation coefficients were calculated between each pair of micronutrients. P-values were corrected for ten pairwise comparisons using the Holm method. Significant correlations were observed between folate and homocysteine (r = − 0.44, p < 0.001), transferrin and vitamin E (r = 0.49, p < 0.001), and folate and vitamin D (r = 0.40, p < 0.001). Additional positive associations were noted between selenium and vitamin B12 (r = 0.34, p = 0.01), and magnesium and vitamin B12 (r = 0.37, p < 0.001), though not all remained significant after correction. Color intensity reflects correlation strength and direction (red = positive, blue = negative).

comparative analysis: vitamin D (25-hydroxyvitamin D), vitamin A (retinol), and homocysteine. Strong correlations were observed between DBS and venous measurements for vitamin A (ρ = 0.74, 95% CI: 0.63–0.81, P < 0.001) (Fig 4a), vitamin D (Spearman ρ = 0.78, 95% CI: 0.69–0.84, P < 0.001) (Fig 4b), and homocysteine (Pearson r = 0.82, 95% CI: 0.76–0.87, P < 0.001) (Fig 4c).

## Discussion

There is still a limited dataset on the fluctuation and deficiency of specific micronutrients during pregnancy and postpartum in the European population. Many studies have been conducted in developing countries, but their findings cannot be directly applied to developed nations due to differences in dietary patterns, healthcare systems, and baseline nutritional status. As a result, there is currently no European consensus on reference values for nutrient levels to accurately identify deficiencies during pregnancy and postpartum. The reference values established for the general adult population may not be appropriate for pregnant women, highlighting the need for pregnancy-specific nutritional guidelines.

**Table 2. Distribution of micronutrient concentration ranges across groups (NP, 1st–3rd, PP).**

| Range | NP | 1st | 2nd | 3rd | PP |
|---|---|---|---|---|---|
| Zn < 11 $\mu$mol/l | 16.67% | 11.11% | 42.86% | 30.77% | 87.5% |
| Zn > 23 $\mu$mol/l | 0% | 0% | 0% | 5.13% | 0% |
| TRF < 2.5 g/l | 16.67% | 5.56% | 4.76% | 0% | 4.17% |
| TRF > 3.8 g/l | 8.33% | 0% | 33.33% | 53.85% | 16.67% |
| HCY > 15 $\mu$mol/l | 8.33% | 0% | 0% | 5.13% | 0% |
| HCY < 5 $\mu$mol/l | 0% | 16.67% | 19.05% | 5.13% | 0% |
| Vit. E > 42 $\mu$mol/l | 8.33% | 0% | 42.86% | 58.97% | 50% |
| Vit. B12 < 156 pmol/l | 20.83% | 22.22% | 28.57% | 56.41% | 50% |
| Se < 0.75 mol/l | 12.5% | 5.56% | 23.81% | 25.64% | 16.67% |
| Vit. A 1.05 − 2.1 $\mu$mol/l | 45.83% | 72.22% | 71.43% | 66.67% | 70.83% |
| Vit. A 2.1 − 4.9 $\mu$mol/l | 54.17% | 22.22% | 23.81% | 15.38% | 29.17% |
| Vit. A 0.65 − 1.05 $\mu$mol/l | 0% | 5.56% | 4.76% | 17.95% | 0% |
| Vit. D < 20 $\mu$g/l | 37.5% | 33.33% | 19.05% | 15.38% | 37.5% |
| Vit. D 21 − 29 $\mu$g/l | 29.17% | 38.89% | 57.14% | 41.03% | 29.17% |
| Vit. D 30 − 44 $\mu$g/l | 33.33% | 27.78% | 23.81% | 38.46% | 16.67% |
| Vit. D 44 − 60 $\mu$g/l | 0% | 0% | 0% | 5.13% | 16.67% |
| Folate < 7.6 nmol/l | 12.5% | 0% | 4.76% | 15.38% | 12.5% |
| Folate 7.6 − 12.2 nmol/l | 33.33% | 11.11% | 28.57% | 10.26% | 8.33% |
| Folate > 12.2 nmol/l | 54.17% | 88.89% | 66.67% | 74.36% | 79.17% |

In this study, we monitored 10 selected nutrients in a Slovak population of pregnant and postpartum women. Levels of certain nutrients remained stable (vitamin D, folate), while others were lower (vitamin A, B12, iron, zinc) or higher (vitamin E) during pregnancy. Notably, a significant proportion of pregnant women exhibited deficiencies in vitamins A, B12, and D, as well as in essential minerals such as zinc and iron. Interestingly, we observed a sharp decrease in homocysteine levels, while vitamin E concentrations exceeded the reference values established for the adult population. These findings suggest distinct physiological adaptations during pregnancy and postpartum that may influence nutrient metabolism and requirements.

Approximately 57% of pregnant women in Europe have 25-hydroxyvitamin D (25(OH)D) levels below 50 nmol/L, indicating widespread vitamin D insufficiency [8]. Despite this, the World Health Organization does not currently recommend routine vitamin D supplementation as part of standard antenatal care [4]. Similarly, an Austrian study reported that 59.5% of pregnant women were vitamin D deficient in the first trimester, 54.8% in the 2nd trimester, 58.5% in the 3rd trimester, and 60% at 12 weeks postpartum, despite nearly 70% of reporting daily intake of a pregnancy supplement containing vitamin D [5]. In a Latvian study, nearly 80% pregnant or postpartum women had inadequate serum vitamin D levels, and dietary intake did not significantly influence their vitamin D status [6]. Finally, a national survey in Belgium revealed that nearly 45% of pregnant women were vitamin D deficient, despite more than 60% reporting the use of multivitamins containing vitamin D during pregnancy [1]. In comparison, our study found a lower prevalence of vitamin D deficiency during pregnancy and the postpartum period (below 50 nmol/L), with 33% of women deficient in the 1st trimester, 19% in the second trimester, 16% in the 3rd trimester, and 38% in the PP period. However, insufficient levels of vitamin D (52.5–72.5 nmol/L) were observed in nearly 60% of the women. This suggests that vitamin D deficiency is highly prevalent among pregnant women in Europe and that multivitamin supplements may not be sufficient to address the issue.

Recent estimates using the probability approach suggest that approximately 82% of pregnant women globally may experience zinc deficiency [9]. However, data on zinc deficiency, particularly within European populations, remain limited.

**a**

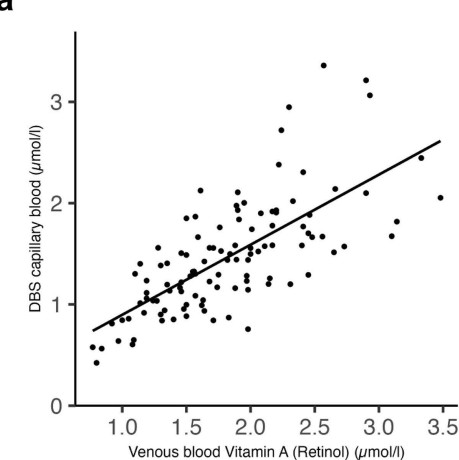

**b**

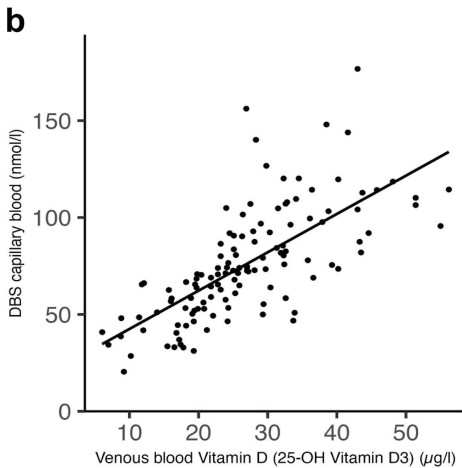

**c**

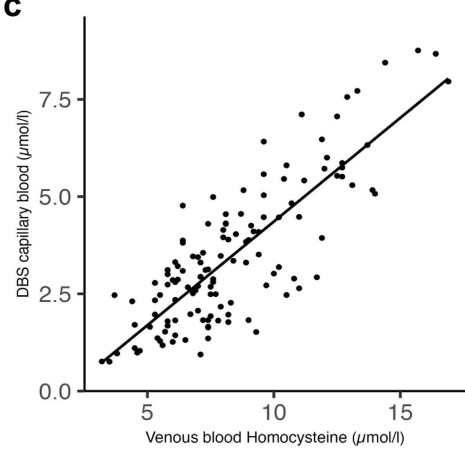

**Fig 4. Correlation between dried blood spot (DBS) capillary blood and venous blood measurements for selected vitamins and homocysteine.**
Scatter plots show paired concentrations of (a) vitamin A (retinol), (b) vitamin D (25-hydroxyvitamin D) and (c) homocysteine measured in DBS capillary blood and venous blood from the same individuals. Solid lines represent linear regression fits. Strong positive correlations were observed for vitamin A ($\rho = 0.74$, $P < 0.001$), vitamin D ($\rho = 0.78$, $P < 0.001$) and homocysteine ($r = 0.82$, $P < 0.001$).

While zinc supplementation during pregnancy is widely practiced, current evidence does not conclusively demonstrate significant improvements in maternal or neonatal outcomes as a result. In our study, 42% of women in their 2nd trimester and 90% of PP women were zinc deficiency. To our knowledge, this is the first comprehensive study indicating that zinc deficiency is highly prevalent among postpartum women and should be closely monitored.

We observed a significant lowering in blood homocysteine levels during pregnancy, dropping to 7–8 µmol/L compared to 12 µmol/L in age-matched NP women. This finding aligns with published data, which report a progressive decrease in plasma homocysteine levels throughout pregnancy (4–6 µmol/L) vs. 8 µmol/L in NP women [10]. This effect has been attributed to folate supplementation and a reduction in albumin concentration. This notion is supported by our data, which show that higher levels of folate are correlated with lower levels of homocysteine. Homocysteine has been identified as an independent risk factor for the development of severe preeclampsia, with elevated plasma homocysteine levels in early pregnancy potentially increasing the risk of non-severe preeclampsia fourfold [11,12]. Given the physiological decrease in homocysteine levels during pregnancy, it is crucial to establish pregnancy-specific reference ranges to enable better monitoring and early detection of hyperhomocysteinemia.

Surprisingly, we observed an excess of vitamin E in the late stages of pregnancy, which may result from excessive intake through food or supplementation. Similarly, Chen et al. (2023) [13] reported that vitamin E levels increased progressively during pregnancy, peaking in late pregnancy, with 15.32% of women exceeding established reference values. It has been proposed that reference values for vitamin E during pregnancy should be higher than those for the general population. A review of pregnancy-related laboratory studies reported trimester-specific reference ranges for vitamin E, showing a progressive increase: 7–13 µg/mL (16.25–30.18 µmol/L) in the 1st trimester, 10–16 µg/mL (23.22–37.15 µmol/L) in the 2nd, and 13–23 µg/mL (30.18–53.40 µmol/L) in the 3rd trimester. The highest concentration range was observed in the 3rd trimester [14]. Gao et al. (2021) established reference ranges for vitamin E levels during pregnancy at 7.4–23.5 mg/L (17.76–56.41 µmol/L) [15]. According to our data, nearly all 3rd trimester pregnant women in our study fall within this range. However, there is currently no European consensus on specific reference ranges for vitamin E blood concentrations in pregnant women. Excessive vitamin E in the 2nd trimester has been identified as a risk factor for gestational diabetes and large-for-gestational-age infants [16]. Additionally, supplementation was associated with an increased risk of abdominal pain and term pre-labour rupture of membranes [17]. Therefore, it is essential to monitor vitamin E levels during pregnancy, and limit supplementation to cases of deficiency.

Routine monitoring of blood nutrient concentrations requires a cost-effective and time-efficient approach that can be easily implemented at home. Therefore, we tested the dried blood spot method to evaluate the precision of nutrient measurement. Our findings revealed a very strong correlation between vitamin A, D, and homocysteine concentrations in venous and capillary blood. This approach enables the development of analytical methods for multiple nutrients identified in our study.

Our study revealed that pregnant women exhibit widespread deficiencies in various vitamins and minerals. It also demonstrated significant differences in micronutrient and homocysteine levels compared to non-pregnant, age-matched women. These findings may prompt discussions on the routine monitoring of selected nutrients during pregnancy and the need for targeted supplementation.

## Limitations

There are several limitations to this study. First, it is not a longitudinal study, which means we could not monitor fluctuations in micronutrient levels over time in individual women. Second, the study was conducted at a single clinical centre in the capital city, which may not reflect the broader population and may limit the generalizability of the findings to rural populations. In fact, we can reasonably expect even higher rates of micronutrient deficiencies in less developed regions of Slovakia, where access to healthcare and nutritional resources may be more constrained. Nevertheless, some of the findings are consistent with other European studies reporting widespread deficiencies in vitamin D and iron.

## Author contributions

**Conceptualization:** Alexandra Kristufkova, Norbert Zilka.

**Data curation:** Alexandra Kristufkova, Norbert Zilka, Katarina Koprdova.

**Formal analysis:** Norbert Zilka.

**Funding acquisition:** Norbert Zilka, Matus Lieskovsky, Michal Fresser.

**Investigation:** Norbert Zilka.

**Methodology:** Neha Basheer, Alexandra Kristufkova, Norbert Zilka, Katarina Koprdova, Matus Lieskovsky.

**Project administration:** Alexandra Kristufkova, Norbert Zilka, Matus Lieskovsky, Michal Fresser.

**Resources:** Norbert Zilka, Matus Lieskovsky.

**Software:** Matus Lieskovsky.

**Supervision:** Alexandra Kristufkova, Norbert Zilka, Matus Lieskovsky, Michal Fresser.

**Validation:** Norbert Zilka.

**Visualization:** Norbert Zilka.

**Writing – original draft:** Neha Basheer, Norbert Zilka.

**Writing – review & editing:** Neha Basheer, Alexandra Kristufkova, Norbert Zilka, Matus Lieskovsky, Michal Fresser.

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
