## [Decision Letter · Decision Letter 0]

18 Jun 2025

PONE-D-25-23187Micronutrient Dynamics and Deficiency Risk Across Pregnancy and Postpartum in a Slovak CohortPLOS ONE

Dear Dr. Zilka,

Thank you for submitting your manuscript to PLOS ONE. After careful consideration, we feel that it has merit but does not fully meet PLOS ONE’s publication criteria as it currently stands. Therefore, we invite you to submit a revised version of the manuscript that addresses the points raised during the review process.

We look forward to receiving your revised manuscript.

Kind regards,

Hansani Madushika Abeywickrama, Ph.D.

Academic Editor

PLOS ONE

Journal Requirements:

We thank Mumo Health for their generous support, which contributed to the execution of this study.

We thank Mumo Health for their generous support, which contributed to the execution of this study.

We thank Mumo Health for their generous support, which contributed to the execution of this study.

Reviewers' comments:

Reviewer's Responses to Questions

**Comments to the Author**

1. Is the manuscript technically sound, and do the data support the conclusions?

Reviewer #1: Yes

Reviewer #2: Yes

2. Has the statistical analysis been performed appropriately and rigorously? 

Reviewer #1: Yes

Reviewer #2: Yes

3. Have the authors made all data underlying the findings in their manuscript fully available?

Reviewer #1: Yes

Reviewer #2: Yes

4. Is the manuscript presented in an intelligible fashion and written in standard English?

Reviewer #1: Yes

Reviewer #2: Yes

5. Review Comments to the Author

Reviewer #1: The authors have produced a well-executed, methodologically sound, and highly relevant study. The manuscript contributes important findings to the field of maternal and public health nutrition, especially within the underrepresented Central European context.

The manuscript is technically sound and presents a rigorously designed cross-sectional cohort study that evaluates the fluctuations in key micronutrients (vitamins A, B6, B12, D, E, zinc, selenium, magnesium, transferrin, and homocysteine) across pregnancy and postpartum in a Slovak cohort.

The data collection methods—using both venous blood and dried blood spot (DBS) sampling—are scientifically appropriate and well-described, and the conclusions are well-supported by the data.

The statistical analysis is robust and suitable for the cross-sectional study design.

With regards to data availability policy, the authors confirm that all data are fully available without restriction. The manuscript explicitly states this and includes detailed tables (e.g., Table 1 and Table 2) listing descriptive statistics, prevalence rates, and reference ranges.

Regarding presentation of the manuscript, the manuscript is clearly written and well-organized, with logical flow between sections.

Overall, the study have some major strengths which include;

a) Comprehensive assessment of 10 micronutrients during pregnancy and postpartum.

b) First-time comparison of DBS and venous blood measurements in a Slovak population.

c) Statistically sound and thoroughly interpreted results.

d) Strong public health implications for antenatal and postpartum care.

However, there are some areas with minor suggestions which include;

a) Consider briefly noting the implications of the study's urban clinical setting and its potential limitation for generalizability to rural populations.

b) A dedicated "Limitations" paragraph would enhance the transparency of the discussion.

c) A minor grammar and syntax check will improve overall readability.

Reviewer #2: This paper is one of the most current papers highlighting the micronutrient deficiency during various important periods as well as the use of DBS as the alternative less-invasive method. This paper is also written well. Some minor improvements are necessary.

1. Line 149, lines 187-188: The subheading is not suitable for the content. The paragraph mentioned the results of all nutrients, not only the nutrients at the subheading. It is suggested to use a more general subheading.

2. Line 426 Table 1: There is no footnote for the two asterisks.

6. PLOS authors have the option to publish the peer review history of their article (what does this mean? ). If published, this will include your full peer review and any attached files.

**Do you want your identity to be public for this peer review?** For information about this choice, including consent withdrawal, please see our Privacy Policy .

Reviewer #1: **Yes: ** Chibuzor Stella Amadi

Reviewer #2: No

---

## [Author Response · Author response to Decision Letter 1]

24 Jun 2025

Reviewers Revisions:

Reviewer #1

1. Consider briefly noting the implications of the study's urban clinical setting and its potential limitation for generalizability to rural populations.

The urban clinical setting's potential impact on generalizability is added to the limitation section

2. A dedicated "Limitations" paragraph would enhance the transparency of the discussion.

We added “Limitations” section to the manuscript.

3. A minor grammar and syntax check will improve overall readability.

We carefully reviewed the manuscript to correct the grammar and syntax.

Reviewer #2

1. Line 149, lines 187-188: The subheading is not suitable for the content. The paragraph mentioned the results of all nutrients, not only the nutrients at the subheading. It is suggested to use a more general subheading.

Line 149. Changed “Vitamin E, Zinc, TRF, and Magnesium Exhibit Stage-Dependent Regulation” to: “Stage-Dependent Variability in Vitamins, Trace Elements, and Homocysteine Levels During Pregnancy and Postpartum”.

Lines 187-188. Changed “High prevalence of vitamin A, D, B12, and zinc deficiencies during pregnancy and postpartum” to: “Prevalence of Micronutrient Deficiencies Across Pregnancy and Postpartum”.

2. Line 426 Table 1: There is no footnote for the two asterisks.

Corrected missing footnote for asterisks in Table 1.

---

## [Decision Letter · Decision Letter 1]

22 Jul 2025

PONE-D-25-23187R1Micronutrient Dynamics and Deficiency Risk Across Pregnancy and Postpartum in a Slovak CohortPLOS ONE

Dear Dr. Zilka,

Thank you for submitting your manuscript to PLOS ONE. After careful consideration, we feel that it has merit but does not fully meet PLOS ONE’s publication criteria as it currently stands. Therefore, we invite you to submit a revised version of the manuscript that addresses the points raised during the review process.

We look forward to receiving your revised manuscript.

Kind regards,

Hansani Madushika Abeywickrama, Ph.D.

Academic Editor

PLOS ONE

Journal Requirements:

Additional Editor Comments:

Please respond to the comments raised by comment 3.

Reviewers' comments:

Reviewer's Responses to Questions

**Comments to the Author**

1. If the authors have adequately addressed your comments raised in a previous round of review and you feel that this manuscript is now acceptable for publication, you may indicate that here to bypass the “Comments to the Author” section, enter your conflict of interest statement in the “Confidential to Editor” section, and submit your "Accept" recommendation.

Reviewer #2: All comments have been addressed

Reviewer #3: (No Response)

2. Is the manuscript technically sound, and do the data support the conclusions?

Reviewer #2: Yes

Reviewer #3: Yes

3. Has the statistical analysis been performed appropriately and rigorously? 

Reviewer #2: Yes

Reviewer #3: Yes

4. Have the authors made all data underlying the findings in their manuscript fully available?

Reviewer #2: Yes

Reviewer #3: Yes

5. Is the manuscript presented in an intelligible fashion and written in standard English?

Reviewer #2: Yes

Reviewer #3: Yes

6. Review Comments to the Author

Reviewer #2: (No Response)

Reviewer #3: The authors have submitted a valuable and timely manuscript that offers important insight into the micronutrient profiles of pregnant and postpartum women in Slovakia. The study is highly relevant to the fields of maternal-fetal medicine, clinical nutrition, and public health. The novelty lies in both the trimester-specific profiling of ten micronutrients and the evaluation of dried blood spot (DBS) sampling as a minimally invasive, scalable diagnostic method.

This research addresses a critical knowledge gap in the European context, where population-based data on pregnancy-specific nutrient dynamics are scarce. The inclusion of age-matched non-pregnant controls, the use of clinically validated assays, and the analysis of both venous and capillary samples contribute to the strength and originality of the manuscript. The findings have the potential to inform prenatal care practices and guide nutritional policy recommendations.

The manuscript is technically sound. It follows a cross-sectional design with clearly defined eligibility criteria and applies a standardized protocol for sampling and nutrient analysis. The statistical approach is appropriate, employing linear regression, Spearman partial correlations, and post hoc testing with Holm corrections. The authors correctly adjusted for multiple comparisons and applied heteroskedasticity consistent variance estimators, which strengthen the validity of their results. The results support the authors’ conclusions that several micronutrients including vitamin A, B12, zinc, and homocysteine exhibit significant physiological variation across pregnancy stages and the postpartum period.

Regarding statistical analysis, all models are appropriate for the study design. However, additional reporting of assumptions (e.g., normality, variance homogeneity) and inclusion of effect sizes and confidence intervals would improve interpretability. Furthermore, it would be helpful to clarify whether key confounders such as BMI, parity, or supplement use were considered in the models, as these may significantly influence nutrient levels. For data availability, the current statement that data are “fully available without restriction” with DOI included complies with PLOS ONE policy.

With respect to presentation, the manuscript is well-organized and written in clear, academic English. The structure adheres to standard scientific conventions. There are a few minor issues with phrasing and grammar (e.g., repetition in “Ethics Committee Ethics Committee” and long, complex sentences in the Discussion section) that should be corrected during final revision.

Suggestions for Improvement:

1.Clearly state whether supplement use, BMI, or parity were included as covariates or excluded from statistical analysis.

2. Conduct a light language edit to address grammatical inconsistencies and remove redundancy.

3. Avoid causal terms like “decline” or “increase” since the study is cross-sectional. Instead, say “lower levels observed in group X” or “group differences.”

In summary, this is a well-designed, clinically relevant study that offers new insight into nutritional challenges during pregnancy in Europe. With modest revisions and improved clarity around data transparency and modeling assumptions, the manuscript will be a strong candidate for publication in PLOS ONE.

7. PLOS authors have the option to publish the peer review history of their article (what does this mean? ). If published, this will include your full peer review and any attached files.

**Do you want your identity to be public for this peer review?** For information about this choice, including consent withdrawal, please see our Privacy Policy .

Reviewer #2: No

Reviewer #3: **Yes: ** Chinwendu Ubani

---

## [Author Response · Author response to Decision Letter 2]

25 Jul 2025

Dear Dr. Abeywickrama,

thank you for the opportunity to revise our manuscript, “Micronutrient Dynamics and Deficiency Risk Across Pregnancy and Postpartum in a Slovak Cohort” [PONE-D-25-23187R1]. We are grateful for Reviewer 3’s constructive feedback and have addressed each point in detail below.

Reviewer Comment 1: Clearly state whether supplement use, BMI, or parity were included as covariates or excluded from statistical analysis.

Response: We appreciate the reviewer’s emphasis on clarifying covariate inclusion. While we collected self-reported data on supplement use during pregnancy, the variability in supplement composition (brand-dependent nutrient content) and inconsistent timing or duration of intake (e.g., folate use only during early pregnancy) precluded its meaningful inclusion as a covariate. BMI data were available for pregnant participants, but not for the non-pregnant control group, making comparisons inconsistent. Moreover, BMI during pregnancy presents interpretive challenges due to gestational weight changes. Parity data were not collected. As such, supplement use, BMI, and parity were excluded from statistical modeling.

This has now been explicitly clarified in the Methods section with the following sentence:

“Supplement use, BMI, and parity were excluded as covariates due to missing or inconsistent data across groups, high variability in supplement intake, and the challenge of standardizing BMI in the context of pregnancy.”

Reviewer Comment 2: Conduct a light language edit to address grammatical inconsistencies and remove redundancy.

Response: We have performed a thorough language review to improve clarity, eliminate grammatical inconsistencies, and remove redundant phrasing. Specific corrections include simplification of long sentences in the Discussion and removal of typographical repetitions such as “Ethics Committee Ethics Committee.”

Reviewer Comment 3: Avoid causal terms like “decline” or “increase” since the study is cross-sectional. Instead, say “lower levels observed in group X” or “group differences.”

Response: We fully agree with this point. The manuscript has been revised to remove causal terminology. Phrases such as “increase” or “decline” have been replaced with observational descriptors like “higher levels observed,” “group differences,” or “variation across groups” to reflect the cross-sectional design of the study accurately.

We believe these revisions address the reviewer’s concerns and have improved the clarity and scientific rigor of the manuscript. We appreciate the reviewer’s thoughtful assessment and hope the revised version meets the journal’s criteria for publication.

Sincerely,

Norbert Zilka,

---

## [Decision Letter · Decision Letter 2]

12 Aug 2025

Micronutrient Dynamics and Deficiency Risk Across Pregnancy and Postpartum in a Slovak Cohort

PONE-D-25-23187R2

Dear Dr. Zilka,

We’re pleased to inform you that your manuscript has been judged scientifically suitable for publication and will be formally accepted for publication once it meets all outstanding technical requirements.

Kind regards,

Hansani Madushika Abeywickrama, Ph.D.

Academic Editor

PLOS ONE

Additional Editor Comments (optional):

Reviewers' comments:

Reviewer's Responses to Questions

**Comments to the Author**

1. If the authors have adequately addressed your comments raised in a previous round of review and you feel that this manuscript is now acceptable for publication, you may indicate that here to bypass the “Comments to the Author” section, enter your conflict of interest statement in the “Confidential to Editor” section, and submit your "Accept" recommendation.

Reviewer #2: All comments have been addressed

Reviewer #3: All comments have been addressed

2. Is the manuscript technically sound, and do the data support the conclusions?

Reviewer #2: (No Response)

Reviewer #3: Yes

3. Has the statistical analysis been performed appropriately and rigorously? 

Reviewer #2: (No Response)

Reviewer #3: (No Response)

4. Have the authors made all data underlying the findings in their manuscript fully available?

Reviewer #2: (No Response)

Reviewer #3: Yes

5. Is the manuscript presented in an intelligible fashion and written in standard English?

Reviewer #2: (No Response)

Reviewer #3: Yes

6. Review Comments to the Author

Reviewer #2: (No Response)

Reviewer #3: The authors have made substantial improvements to the manuscript, resulting in a well-structured and scientifically sound paper. The study design and rationale are now communicated with greater clarity, and the data analysis methods have been presented in a manner that demonstrates rigor and appropriateness for the research objectives.

A key strength of this revised manuscript lies in its transparent explanation for the exclusion of certain variables specifically BMI, supplement use, and parity. The justification provided is both logical and clearly articulated, reflecting thoughtful consideration of data quality, consistency, and methodological relevance. This enhances the credibility of the analysis.

The authors have also appropriately revised the language throughout the manuscript to remove any causal implications, adopting observational descriptors that align with the cross-sectional nature of the study. This shift improves the scientific integrity and interpretability of the findings.

Importantly, the justification for the use of dried blood spot (DBS) sampling is compelling. The authors explain its practicality, accessibility, and comparability to venous sampling with clarity, supporting its relevance for both clinical settings and population-level monitoring. This methodological approach significantly strengthens the study’s clinical applicability.

Finally, the manuscript addresses a critical data gap in maternal nutrition surveillance in Central Europe. The findings contribute valuable insight into micronutrient trends during pregnancy and postpartum an area with limited region-specific evidence. The work stands to inform future public health strategies and nutritional interventions in similar contexts.

Overall, the manuscript is now well-prepared for publication and represents a meaningful contribution to maternal health research.

7. PLOS authors have the option to publish the peer review history of their article (what does this mean? ). If published, this will include your full peer review and any attached files.

**Do you want your identity to be public for this peer review?** For information about this choice, including consent withdrawal, please see our Privacy Policy .

Reviewer #2: No

Reviewer #3: **Yes: ** Chinwendu Ubani

---

## [Editor Report · Acceptance letter]

PONE-D-25-23187R2

PLOS ONE

Dear Dr. Zilka,

I'm pleased to inform you that your manuscript has been deemed suitable for publication in PLOS ONE. Congratulations! Your manuscript is now being handed over to our production team.

Kind regards,

on behalf of

Dr. Hansani Madushika Abeywickrama

Academic Editor

PLOS ONE